# Development and Application of Dual-Sensors Label in Combination with Active Chitosan-Based Coating Incorporating Yarrow Essential Oil for Freshness Monitoring and Shelf-Life Extension of Chicken Fillet

**DOI:** 10.3390/foods11213533

**Published:** 2022-11-06

**Authors:** Seyed Hadi Peighambardoust, Milad Yaghoubi, Azam Hosseinpour, Kazem Alirezalu, Maral Soltanzadeh, Mohammadreza Dadpour

**Affiliations:** 1Department of Food Science, College of Agriculture, University of Tabriz, Tabriz 5166616471, Iran; 2Department of Food Science and Technology, Ahar Faculty of Agriculture and Natural Resources, University of Tabriz, Tabriz 5166616471, Iran; 3Department of Horticultural Sciences, College of Agriculture, University of Tabriz, Tabriz 5166616471, Iran

**Keywords:** chicken meat, active coating, freshness sensing, spoilage, intelligent packaging, shelf life

## Abstract

This study aimed for the application of active chitosan coating incorporating yarrow essential oil (YEO) together with the development of an on-package sensor label based on bromocresol purple (BCP) and methyl red (MR) for shelf-life extension and freshness monitoring of chicken breast fillet. Physiochemical and microbiological attributes of chicken meat coated with sole chitosan, YEO, and chitosan + YEO were compared with those of uncoated (control) samples. Chitosan + YEO coated chicken meat stayed fresh with no significant changes (*p* > 0.05) in pH (5.42–5.56), TVB-N (12.55–15.36 mg N/100 g), TBARs (0.35–0.40 mg MDA/kg) and total aerobic psycrotrophic bacteria (3.97–4.65 log CFU/g) in days 1–15. There was no response of the dual-sensors label toward the variation in chemical and microbiological indicators of chicken meat coated with chitosan + YEO. However, either uncoated, sole chitosan, or sole YEO treatments indicated a three-stage freshness status with the fresh stage belonged to a period earlier than day 7 (with no distinct color change in both sensor labels); the semi-fresh stage corresponded to storage days between 7–9, wherein a gradual color change appeared (MR from pink to orange, BCP from yellow to light purple); and the spoiled stage occurred in day 9 onward with a drastic color change (MR from orange to light yellow, BCP from light purple to deep purple). In general, the dual-sensors successfully responded to the variation of chemical and microbiological indicators and visual color of uncoated samples during storage time. Based on the obtained results, the application of chitosan + YEO coating efficiently prolonged the freshness of chicken breast meat, where on-package dual-sensors systems were able to detect the freshness stages of meat samples during storage time.

## 1. Introduction

Due to high moisture content and neutral pH (5.5–5.6), fresh chicken meat is highly perishable and susceptible to microbial growth, and lipid and protein oxidations lead to the development of off-flavor and short shelf-life [1]. Therefore, reliable techniques for preserving and assessing the freshness of meat would be beneficial. The application of active natural coatings incorporating essential oils and packaging equipped with freshness indicators is one of the most practical and emerging techniques in simultaneous monitoring/improving the stability and freshness of poultry meat products.

Generally, edible coatings are produced from polysaccharides, lipids, and proteins utilized in combination with each other, or solely [2,3]. Chitosan, which is produced commercially from shellfish processing waste, is a cationic polysaccharide obtained by the deacetylation of chitin [4,5]. Chitosan with its practical characteristics such as film-forming ability, antimicrobial and antioxidant activity, good barrier properties, non-toxicity, biodegradability, and biocompatibility properties can be effectively used for food preservation [6,7]. The efficacy of chitosan edible coatings to delay putrefaction and quality deterioration of foods has been indicated [8,9]. In this regard, the potential antioxidant and antimicrobial activities of chitosan coatings [10] combined with other natural additives have been demonstrated in pork slices [11], chicken breast meat [12], refrigerated pork [13], and beef [14]. Yarrow (*Achillea millefolium* L.) is cultivated widely in America, Europe, and Asia and is one of the most important medicinal plants used extensively in the food, cosmetic, and pharmaceutical sectors [15,16]. The potential anti-inflammatory, antimicrobial, and antioxidant activities of YEO have been reported [17]. According to Farhadi et al. [18], YEO is a rich source of campherol, lutein, rutin, and apigenin with techno-functional properties.

The interest and demand for fresh and safe packaged foods are growing nowadays [19]. Intelligent packaging appeared to meet this requirement by monitoring the package conditions and informing consumers about the freshness of the packaged food [20]. For predicting and monitoring meat products’ shelf-life and monitoring the spoilage stages, intelligent packaging has been widely utilized. However, it is still a challenge to develop cost-effective, easily accessible, and simple techniques for fast and real-time evaluation of meat spoilage. Freshness indicators as one of the elements of intelligent packaging systems are able to monitor and inform freshness/spoilage conditions in a real-time manner [21]. The freshness sensors are typically halochromic indicators that give consumers important information about chemical and microbial spoilage status of packaged products [22]. These sensors can monitor gas molecules, such as CH_4_, NH_3_, H_2_S, and CO_2_, along with total volatile basic nitrogen (TVB-N) and volatile organic compounds related to enzymatic food spoilage. Indicator dyes such as methyl red (MR) with pKa of 5.1 and bromocresol purple (BCP) with pKa of 6.3 could also be used as freshness indicators for poultry meat. MR appears red in pH < 4.4, orange in 4.4 < pH > 6.2, and yellow in pH > 6.2. BCP is colored yellow below pH 5.2, and violet above pH 6.8. Since fresh chicken meat has a pH of around 5.50, increasing pH above 6.0 as a function of spoilage chemical reaction (i.e., the liberation of volatile amines) can easily lead to color changes in the indicator dyes [23]. Therefore, the interaction among colorimetric indicators and volatile alkaline components produced in the headspaces of packaged meat samples leads to changes in indicator color that are easily detectable by the naked eye as known as a simple, economic and practical technique.

The pH indicators potentially can be utilized for the detection of microbial balance and monitoring the meat freshness as an on-package label. In general, a pH sensing indicator is fabricated based on a solid support and a dye with the ability to react to the chemical substances originating from food spoilage. There are different synthetic dyes used in the preparation of freshness indicators, among which the most commons are: methyl red, bromocresol green, bromocresol purple, cresol red, chlorophenol, bromothymol blue, and xylenol [24]. Absorption is one of the practical and effective methods to deposit the dye on a solid support with a high absorption capacity. In this process, the support is immersed in a colorant solution and there should be a strong affinity between the solid support and the indicator dye leading to low and controlled leaching behavior over time. In this regard, cellulose-based supports such as filter paper, have been extensively used for the fabrication of freshness sensors by the absorption method due to their high water-holding capacity [25,26].

The application of a single MR-based pH indicator has been reported for real-time monitoring of broiler chicken cut freshness [23]. In a later study, a dual-sensors label based on pH indicators for real-time monitoring of beef freshness was introduced [26]. Additionally, in a recent study, the capability of intelligent pH-sensitive colorimetric labels by immobilizing anthocyanins into bacterial nanocellulose support has been studied for monitoring the freshness status of shrimp during cold storage [27]. To the best of our knowledge, the combined effect of active chitosan coating incorporating YEO and application of freshness dual-sensors label based on MR and BCP on quality characteristics and shelf-life of chicken breast fillet has not been reported yet. Therefore, the main objective of this work is to develop intelligent on-package dual-sensor labels for real-time monitoring of the freshness/spoilage of uncoated chicken breast meat and those coated with sole chitosan, YEO, or chitosan + YEO coating during the chilled storage period.

## 2. Materials and Methods

### 2.1. Chemicals

All chemicals used were of analytical grade supplied by Merck (Darmstadt, Germany) or Sigma-Aldrich (St. Louis, MO, USA). Stock solutions of methyl red (MR) and bromocresol purple (BCP) were prepared by dissolving 10 mg of each substance in 10 mL of ethanol (50%) to give a concentration of 1 mg·mL^−1^.

### 2.2. Fabrication of Dual-Sensors Label by Adsorption Method

To prepare a dual-sensors label an absorption method of Kuswandi and Nurfawaidi [26] was used with slight alteration described below. Briefly, the Whatman filter paper (no. 1001) sheets were dipped in the stock solutions of either MR or BCP (1 mg·mL^−1^) for 12 h at room temperature. To remove unbound dye within the paper labels, they were gradually washed out with distilled water using a Piset. The papers were then immediately dried with an electrical drier. The dried filter papers were then cut into similar circular shapes (inner and outer circles correspond to BCP and MR labels, respectively) with a scissor. Afterward, both MR and BCP circular membranes were center-joined together using a double-sided tape to design an on-package dual-sensors label shown in Figure 1. It is worth mentioning that the junction of completely dried papers with a double-sided tape would avoid any dye diffusion between the filter paper membranes during storage days. In this case, both membranes were separated and not tightly bound to each other. Finally, the dual-sensors label was attached to the inner side of the packaging film by double-sided tape.

### 2.3. Essential Oil Extraction

The Yarrow is mostly grown in southern provinces of Iran [28]. Fresh Yarrow collected in Autumn 2021 was prepared from Pergas-Giyah farms of Shiraz (Fars, Iran) and was authenticated by the botanical herbarium of the University of Tabriz (Tabriz, Iran). The collected aerial parts were washed, cut into small pieces, and shade-dried at room temperature (25 °C). Extraction of essential oil (EO) content was carried out using the steam-distillation method by a Clevenger-type apparatus (Schott Duran, Mainz, Germany) for 6 h. The isolated EO was transferred to a glass vial and was frozen at −20 °C to separate the entrapped water droplets as an ice phase [7]. YEO was then drained from the vial and kept at −20 °C in dark until the next use.

### 2.4. Gas Chromatography (GC)–Mass Spectrometer Analysis

The chemical profile of YEO was characterized by a GC (GC-17A, Shimadzu Inc., Kyoto, Japan) coupled to a mass spectrometer (Model QP-5050A) according to the detailed procedure reported earlier [7]. The components in GC-MS spectra were reported based on their retention time (*RT*), retention (kovats) index (*RI*), and integral areas (A%). *RI* was calculated based on obtained *RT* of each component relative to *RT* of a series of C_5_–C_24_ n-alkanes on a DB-5 column according to Equation (1):(1)$$RI=100\times [n+\left(N-n\right)\times \left(\frac{Log\;RT_{unknown}-Log\;RT_{n}}{Log\;RT_{N}-Log\;RT_{n}}\right)$$
where *n* and *N* represent the numbers of carbon atoms in the smaller and larger n-alkanes, respectively. *RT* is the retention time of the related compounds.

The identification of components was carried out by comparing mass spectral peaks and RI with those available in the literature [29] as well as by computer matching using the National Institute of Standards and Technology (NIST 21 and 107) and WILEY (229) mass spectral libraries.

### 2.5. Preparation of Chicken Breast Fillets, Application of Active Coatings and Packaging

Fresh skinless chicken breast (pH 5.5–5.6, 2 h after slaughter) was purchased at a local poultry meat shop (Azar-morgh, Ahar, Iran) and directly transported to the laboratory within 30 min at cold conditions. Moisture, protein, fat, and pH values of chicken breast samples were 76.12 ± 1.56, 20.91 ± 0.90, 1.5 ± 0.09, and 5.85 ± 0.19, respectively. Chicken breasts were sampled into portions of almost equal weight (50 ± 5 g) by a sterile knife, covered with low-permeability polyethylene plastic films, and kept refrigerated before coating treatments. To prepare coating solutions medium Mw chitosan powder (20 g) with a deacetylation degree of 95% was dissolved in 1% acetic acid and reached 1000 mL to give concentrations of 2% *w*/*v*. The obtained solution was filtered twice and 75 mL of glycerol as a plasticizer and 16 g Tween 80 as an emulsifier were added, followed by stirring for 1 h at room temperature. YEO at concentrations of 0.3% *v*/*v* was added to the above solution and homogenized using an Ultra-turrax (T25 IKA Werke, Germany) at 20,000 rpm for 3 min. Finally, the coating solutions were filtered and degassed. Chicken fillets were dipped in the latter solution for 10 min at 4 °C. The excess solution was drained off immediately after dipping and the fillets were allowed to dry. For a comparison purpose of the efficacy of chitosan versus YEO, a coating treatment was also prepared by direct addition of YEO on chicken breast fillets without the application of chitosan coating. For this purpose, an emulsion of deionized water, Tween 80, and neat YEO at 0.3% *v*/*v* was prepared. The direct addition of YEO simulated the previous three steps of dipping—draining—drying as explained for chitosan-incorporated coatings preparation. Thus, four treatments were prepared as: T1 (Ch 0%—YEO 0%), T2 (Ch 2%—YEO 0%), T3 (Ch 0%—YEO 0.3%), and T4 (Ch 2%—YEO 0.3%). The control (T1) was the chicken slice dipped into sterile distilled water with no coating material. Prepared chicken fillets in similar weights were placed inside zip-lock low-density (0.9 g·cm^−3^) polypropylene bags. Then, the dual-sensors stickers were positioned inside the packaging atmosphere and attached by a small double-sided tape to the inner surface of the film, while keeping a suitable distance close to meat samples to ensure avoiding direct contact with the meat surface. The packaged meat samples were stored under chilled conditions (4 ± 0.2 °C). The physicochemical, microbiological, and color attributes of meat samples were assessed at 1, 3, 5, 7, 9, 12, and 15 days of storage period. At the same time, the color response of dual-sensors stickers was also assessed.

### 2.6. Measurement of TBARs, TVB-N, and pH

A spectrophotometer (Hitachi, Tokyo, Japan) at 532 nm was used for measuring the components as a result of the reaction between thiobarbituric acid and chicken breast oxidation products expressed as thiobarbituric acid reactive substances (TBARs). The TBARS assay measures malondialdehyde (MDA), which is the degradation product of lipid oxidation [30]. The standard curve was obtained using 1,1,3,3-tetraethoxypropane at 0–10 ppm concentrations, and finally, the results were expressed as mg MDA/kg of samples [31]. The total volatile basic nitrogen (TVB-N) is one of the traits used as an indicator of freshness, with higher T-VBN values indicating that chicken breast meat is less fresh [30]. The TVB-N of meat samples was assessed by the Kjeldahl apparatus with a vapor distillation according to the method described by Qiao et al. [32]. The data were expressed as mg N/100 g chicken meat samples. The pH of the samples was assessed using a calibrated pH meter (Methrom, Herisau, Switzerland) after mixing 10 g of samples with 100 mL of distilled water.

### 2.7. Microbiological Analysis

The effect of active coatings on the microbial count of chicken breast fillets stored under refrigerated conditions was evaluated on days 0, 3, 5, 7, 9, 12, and 15th of storage periods. Microbiological analysis was concentrated on the main spoiling groups of microorganisms in the chicken breast: Total aerobic mesophilic bacteria (AMB), total aerobic psycrotrophic bacteria (APB), *Enterobacteriaceae* (coliforms), and yeasts and molds [33]. Microbial analysis was performed according to the method of Fernández-Pan et al. [33] with slight alteration described below. Briefly, 10 g of chicken meat samples were placed in a sterile plastic bag with 90 mL of sterile peptone water (0.1% *w*/*v*) and homogenized in a stomacher (Bagmixer 400 W, Interscience, St Nom, France) for 5 min. Serial decimal dilutions were then prepared using 1 mL of the obtained suspension with 9 mL of sterile peptone water (0.1% *w*/*v*). An aliquot of 0.1 mL of the chicken homogenates was seeded over specific agar media corresponding to each group of bacteria to be analyzed. Applied conditions for each microbiological analysis are shown in Table 1. All results were reported as logarithms of the number of colony-forming units (Log CFU) per g of chicken meat [34].

### 2.8. Determination of Color Parameters

The color attributes of chicken samples were analyzed using a digital image processing method according to a method of León et al. [35] with the alteration described below. Digital images of meat samples were captured in a photography box (50 × 50 × 25 cm; with two 25 W LED lamps on the top with a radiation angle of 90 degrees) using a digital camera (Canon SX720, Tokyo, Japan) mounted on the top center of the box. The acquired images were processed using Adobe Photoshop 21.1.3 (Adobe Systems Inc., Salinas, CA, USA) with a ‘*L a b* color’ image mode to obtain *L** *a** *b** values at six different locations [9]. Color parameters were then calculated using the following equations [36,37]:(2)$$C^{*}=\sqrt{{\left(a^{*}\right)}^{2}+{\left(b^{*}\right)}^{2}}$$
(3)$$WI=100-\sqrt{{\left(100-L^{*}\right)}^{2}+{\left(a^{*}\right)}^{2}+{\left(b^{*}\right)}^{2}}$$
(4)$$YI=\frac{142.86\times b^{*}}{L^{*}}$$
where *L** = lightness, *a** = redness/greenness, *b** = yellowness/blueness, *C**, *WI* and *YI* are chroma, whiteness, and yellowness indices of the samples, respectively [38,39].

### 2.9. Measurement of the Response of Dual-Sensors Label

The packages containing different samples were stored under chilled conditions for different storage days in order to evaluate the applicability of the dual-sensors to monitor the spoilage process of the chicken breast fillets. The distinct irreversible color change of the MR sensor from pink to yellow and that of the PCB sensor from yellow to purple was used as the measurable response of change [23,26]. Digital images of the dual-sensors were captured as explained in Section 2.8. The images were analyzed using Adobe Photoshop 21.1.3 (Adobe Systems Inc., Salinas, CA, USA) with an ‘RGB color’ image mode. At least three selections were taken from each image and averaged via the “Filter—Blur—Average” function of a software to give RGB values. Then an average of each *R*, *G*, and *B* was calculated. To get the “*mean RGB*” values for each color sensor, the following equation was set:(5)$$mean\;RGB=\sqrt{\left(\frac{R^{2}+G^{2}+B^{2}}{3}\right)}$$

### 2.10. Statistical Analysis

All data were statistically analyzed under a completely randomized design (CRD) framework subjected to the analysis of variance (ANOVA) using Statistical Analysis System software (SAS version 9.2, SAS Inc., Cary, NC, USA). A one-way analysis of variance (ANOVA) and Tukey’s multiple range test (at *p* < 0.05) were used [40]. Triplicate measurements were performed for all analyses, except for those mentioned in the methods.

## 3. Results

### 3.1. Gas Chromatography-Mass Spectrometry Analysis

The aerial parts of the Yarrow plant (*Achillea millefolium*) gave an average essential oil yield of 1.7 ± 0.2% (dry weight basis). The chemical composition of YEO broadly varies with geobotanical conditions, harvesting season, herbal maturity, and extraction procedure [16,18]. GC-MS chromatogram and the volatile chemical compounds of YEO analyzed in this study are presented in Figure 2 and Table 2, respectively. A total of twenty-seven chemicals were identified accounting for 94.7% of the total EO composition. Oxygenated monoterpenes were the most predominant compounds, which were characterized by camphor (18.6%), eucalyptol (17.1%), and α-thujone (12.1%). In total, these terpenoids comprised almost half (47.8%) of the compounds of YEO. Other important constituents (as a total of 22.5%) were also characterized as α- terpineol (7.2%), citral (neral + geranial, geranyl acetate) (6.1%), β-thujone (3.8%), camphene (3.0), and limonene oxide (2.4%).

Chemical analysis of YEO by GC-MS reported in other studies supports our results indicating that camphor, caryophyllene, eucalyptol, and borneol are the most prevalent components of YEO [16,18,28,41]. Eucalyptol is a monoterpenoid providing a fresh mint- and camphor-like smell and a spicy, cooling taste in YEO. The name of this component relates to eucalyptus which comprises up to 90% of the EO of *Eucalyptus polybractea*. Eucalyptol and camphor, as the main components of YEO, characterized in this study, are responsible for bioactive functions such as antioxidant, antimicrobial, insecticide, antibacterial, and fungicide effects [42]. The α-thujone also provides a minty aroma and it has been shown to exhibit medicinal benefits for inflammatory disorders and to provide strong anti-fungal function [43].

### 3.2. Effect of Coating Type and Storage Time on pH, Thiobarbituric Acid Reactive Substances (TBARs), Total Volatile Basic Nitrogen (TVB-N) Values of Chicken Breast Fillet

#### 3.2.1. pH

Variation of pH values of chicken fillets as a function of both coating type and refrigerated storage time is given in Table 3. The pH values of the control (uncoated) samples were increased from 5.86 to 7.26. There was no significant difference (*p* > 0.05) between pH values before day 7. This day specifically presented an onset for starting a significant pH increase. The highest pH value of 7.26 was obtained for uncoated samples on day 15. The increase in the pH of meat upon storage can potentially arise from its microbiological status. According to Radha Krishnan et al. [44] and Alirezalu et al. [45], increasing in meat pH might be attributed to an accumulation of alkaline compounds as a result of psycrotrophic bacteria activity and autochthonous enzymes’ autolytic activity. The increase in pH was remarkably higher in the uncoated samples. A similar trend was observed for chicken fillets treated with sole YEO (Ch_0%—YEO_0.3%) indicating that application of YEO alone cannot stop pH rise during storage. EO can be easily decomposed in its free form when applied on a meat surface without coating. Therefore, chitosan-coated samples without YEO (Ch_2%—YEO_0%) showed a similar trend to that of control, but at lower initial pH values due to the acidic nature of chitosan. Meat samples coated with Ch_2%—YEO_0.3%, however, showed no significant changes (*p* > 0.05) in their pH values during storage indicating no sign of spoilage. Sujiwo et al. [30] reported that the predominant spoilage microorganisms in chicken meat surfaces can only grow at higher pH ranges (pH > 5.8).

#### 3.2.2. TBARs

In meat and meat products, the quality attributes and shelf-life can be negatively affected by oxidation reactions. The secondary products of oxidation, particularly aldehydes, can be measured using the TBARs value as an indicator of meat freshness [46]. The variation of TBARs values as a function of both coating type and refrigerated storage time is given in Table 3. The TBARs of the uncoated sample (control) gradually and non-significantly (*p* > 0.05) increased from 0.34 to 0.45 mg MDA/kg on days 1–7. Similar to pH values, day 7 presented an onset of remarkable TBARs change, and day 9 was attributed to a sharp and significant (*p* < 0.05) TBARs increase indicating a threshold of spoilage in the control sample. A similar trend was observed for Ch_2%—YEO_0% and Ch_0%—YEO_0.3% coatings, indicating that the application of either chitosan or EO alone did stop TBARs production upon storage time. However, for the Ch_2%—YEO_0.3% coated sample, there were no significant changes (*p* > 0.05) in TBARs on days 1–12, and the values were within the range of acceptable values (below 0.40 mg MDA/kg). It is reported that TBARS values below 0.40 mg MDA/kg should be acceptable for consumers indicating fresh meat, and the values in the range of 0.6 to 2.0 mg MDA/kg could produce off-odors and off-flavors that can be detected by inexperienced consumers [30]. The variation of TBARs values can explain pH changes in chicken meat samples. The results of the present study are in parallel with those obtained by Yaghoubi et al. [46] who showed the impact of chitosan coating incorporating *Artemisia fragrans* essential oil on chicken fillet quality upon storage. Jonaidi Jafari et al. [47] also reported similar results in chicken fillets treated with chitosan-containing propolis extract. These authors observed a less increase in TBARs values in coated samples (<0.6 mg MDA/kg) compared to uncoated samples (>0.8 mg MDA/kg).

#### 3.2.3. TVB-N

As one of the most vital indicators in the shelf-life of meat and meat products, TVB-N mainly includes amines and ammonia [48]. Generally, in spoiled chicken meat, the volatile basic amine (TVB-N) levels increase because of the formation of NH3 along with other volatile and biogenic amines [49]. According to Table 3, The TVB-N values of control meat increased from 12.91 to 35.20 mg N/100 g. Similar to pH and TBARs values, TVB-N was significantly (*p* < 0.05) increased on day 9 as the threshold of remarkable chemical changes in volatile nitrogen showed the spoilage stage. Ch_2%—YEO_0% and Ch_0%—YEO_0.3% coating treatments exhibited a similar trend to that observed for the control sample indicating that application of either chitosan or EO alone did not stop TVB-N change upon storage time. However, Ch_2%—YEO_0.3% coated sample showed no significant changes (*p* > 0.05) in TVB-N on days 1–12 and the values 11.60–19.36 mg N/100 g were within the permitted range. It has been reported that 25 mg/100 g is the permitted level of TVB-N values related to microbiological contamination and freshness loss of meat and meat products. Similar results were reported by Yaghoubi et al. [46] and Mojaddar Langroodi et al. [50] who indicated that chitosan-based coatings containing natural preservatives reduced TVB-N values in meat. The microbiological results and changes in pH and TBARs values agree with TVB-N values since the strong antimicrobial and antioxidant attributes of chitosan coating incorporating YEO could be responsible for lower pH and TBARs values as well as lower microbiological count.

### 3.3. Microbiological Analysis of Chicken Samples

The effects of different coating treatments on the viability of aerobic mesophilic bacteria (AMB), aerobic psycrotrophic bacteria (APB); coliforms (*Enterobacteriaceae*), and yeast and molds are presented in Table 4.

The population of all types of above-mentioned microorganisms in the control sample (with no coating) showed a steady non-significant (*p* > 0.05) increase from day 1 to 5, but a sharp and significant (*p* < 0.05) increase from days 7–9 onwards. In the control sample, days 7–9 presented a threshold of remarkable change in the microbiological state of meat indicating the start of spoilage. A similar trend was observed for Ch_2%—YEO_0% and Ch_0%—YEO_0.3% coatings indicating that application of either chitosan or EO alone did not cease microbial growth from day 9 onwards. However, the Ch_2%—YEO_0.3% coated sample showed no significant changes (*p* > 0.05) in colony-forming units of all microorganisms in all days of storage. The electrostatic interaction among microbial cell membranes (as a negative charge) and the chitosan NH_3_ group (positive charge) gives rise to the leakage of intracellular compounds [7]. Furthermore, as reported in the literature, a combination of the anti-microbial activity of plant EO with that of chitosan provides a synergistic effect on the anti-microbial activity of the resulting coating [9]. In fresh meat, the acceptable limitation for total aerobic mesophilic bacteria count is 6 log CFU/g [51]. The control samples exceeded the acceptable microbial limitation on day 9, while a combination of chitosan and YEO revealed acceptable levels until day 15. The variation in the microbiological state of samples is in parallel with changes in pH, TBARs, and TVB-N values indicating that those changes were probably initiated by the growth and activity of psycrotrophic bacteria in chicken meat samples upon refrigerated storage time. It can be concluded that the visual color change of the dual-sensors during storage days is a useful indicator for the assessment of microbial spoilage status in chicken meat. Similar results were reported in chicken meat coated with chitosan enriched with *Zataria multiflora* EO [52], *Artemisia fragrans* EO [46], and propolis extract [47]. According to Pabast et al. [53], lamb meat coated with chitosan containing nano-encapsulated *Satureja* plant EO exhibited extended shelf-life during storage.

### 3.4. Chicken Meat Color

As one of the most important parameters in meat and meat products, color attributes can potentially affect consumer acceptance [54]. Variations in the visual color of chicken meat treated with different types of coatings along with their dual-sensor labels color change as a function of storage time are shown in Figure 3. As can be seen from this figure, all samples exhibited the fresh color of chicken breast fillet in the early days of storage (before day 5). The color of chicken meat started to moderately change on day 7, but a very distinct change in the color of control, Ch_2%—YEO_0% and Ch_0%—YEO_0.3% samples occurred from day 9 onwards. However, the visual color of the Ch_2%—YEO_0.3% sample was not remarkably affected during all days of storage. To better understand and quantify the color changes in all meat samples, color indices in terms of chroma (C*), whiteness index (WI), and yellowness index (YI) were investigated as a function of coating type and storage time. These indices were obtained from calculated CIE Lab values of each sample according to Equations (2)–(4).

Table 5 represents a variation of C*, WI, and YI indices in different chicken meat treatments. The C* of control decreased from 28.80 on day 1 to 15.71 on day 15. The significant change in C* value appeared in an interval of day 7–day 9. A similar trend was observed for Ch_2%—YEO_0% and Ch_0%—YEO_0.3% treatments, but Ch_2%—YEO_0.3% sample did not show significant changes (*p* > 0.05) in the C* index throughout storage days. The latter treatment successively preserved chicken meat color during the storage period. Chroma is the amount of saturation or pureness of a color. For example, colors with a high C* are said to be clear or bright. Whereas, dull colors exhibit low C* values [55].

Table 5 shows that C* varied in parallel with chemical and microbial changes occurred in meat samples indicating that fresh and spoiled meats exhibited the highest and lowest C* values, respectively. WI and YI indices represent the amount of white and yellow color development in meat samples, respectively. These indices are preference ratings for how white or yellow a material should appear. Fresh uncoated (control) chicken meat samples (days 1–3) exhibited higher YI and lower WI values compared to semi-fresh (days 5–7) and spoiled (days 9–15) samples. A similar trend was observed for Ch_2%—YEO_0% and Ch_0%—YEO_0.3% treatments, but Ch_2%—YEO_0.3% treatment did not show significant changes (*p* > 0.05) in the WI and YI and C* indices throughout storage days. The antimicrobial and antioxidant attributes of chitosan together with YEO may be the main reason for high C* and YI values in the latter treated samples. Reversely, the reduction in C* and YI indices of uncoated meat samples during storage might be related to the liberation of free radicals originating from either lipid oxidation or met-myoglobin formation. The yellowness of meat can be affected by oxidative or enzymatic reactions throughout the storage period [56].

### 3.5. Response of Dual-Sensors in Relation to Different Stages of Chicken Meat Freshness

For safety reasons, dual MR/BSC stickers are not allowed to come into contact with meat samples. In this study, these stickers were attached to the inner surface of zip-lock polyethylene bags by keeping a suitable distance close to meat samples, in order to respond to the increase of volatile basic amines originating from chicken spoilage reactions. The spoilage will be specified by a distinct color change of the MR sensor from pink to yellow, and the PCB sensor from yellow to purple. As an advantage of the dual-sensors system, the false positive or negative response can be avoided, since both sensors were referenced each other, avoiding individual sensors going false. The false negative could only occur if the plastic cover of the package is broken, liberating volatile amines produced during meat deterioration, reducing their concentration inside the package, which in turn can cause a false sensor response.

The variation of pH values of chicken meat as a function of both coating type and refrigerated storage time was shown in Table 3. Dual-sensors were sensitive to pH changes in uncoated meat samples, thus, the visual color, as well as mean RGB values of both MR and BCP sensors, were plotted against pH and the results are shown in Figure 4. There was a color change in the MR label from pink to yellow and in the BCP sensor from yellow to purple by an increase in meat pH. This color response in both sensors was useful to monitor the freshness stages of chicken meat during storage enabling this technique to evaluate the functionality of different coatings types on meat spoilage prevention.

The rate of the color change of dual-sensors toward the spoilage of chicken meat as a function of different treatments (coating with/without EO incorporation) and storage time shown by mean RGB values is demonstrated in Figure 5. Four coating treatments (Figure 5A–D) have been compared in regards to dual-sensors color responses during storage days. The most remarkable color changes were seen for the uncoated (control) sample (Figure 5A) and no distinct color change was observed for treatment with chitosan + YEO (Figure 4D). Treatments with no coating (Ch_0%—YEO_0%, Figure 5A), only chitosan coating (Ch_2%—YEO_0%, Figure 4B), and only YEO (Ch_0%—YEO_0.3%, Figure 5C) showed a three-stages color change corresponding to different meat freshness status, where the fresh stage belonged to a storage period earlier than day 7 (with no distinct color change in both sensor labels); the semi-fresh stage corresponded to storage days 7–9, wherein a gradual color change appeared in sensor labels (MR changed from pink to orange, BCP changed from yellow to light purple); and the spoiled stage occurred in day 9 onward with a drastic and increasing color change (MR changed from orange to light yellow, BCP changed from light purple to deep purple). It is worth mentioning that in coating treatment with chitosan incorporating YEO (Ch_2%—YEO_0.3%, Figure 5D), there was no color change of both sensors in all storage days indicating that chicken meat stayed fresh over the storage period. Previous visual inspection of the color of both meat and corresponding sensor labels (Figure 3) did not detect differences in the color of the Ch_2%—YEO_0.3% sample as a function of storage days. According to sensor color response, the onset of chicken meat spoilage was detected in a semi-fresh stage on days 7–9, followed by a threshold of spoilage on day 9. Since no more color change was seen after this period and the meat underwent spoiling conditions from day 9 onward.

### 3.6. Response of Dual-Sensors to Variation of pH, TVB-N and TBARs

Chemical indicators in relation to the type of coating applied and storage time were explained earlier in Section 3.2 (Table 3). Here, to evaluate how dual-sensors labels respond to these chemical indicators, variations of pH, TVB-N and TBARs were plotted against mean RGB values (Figure 6A–C).

Figure 6(A1–C1) and Figure 6(A2–C2) demonstrate the variation of pH, TVB-N, and TBARs in uncoated (control) and Ch_2%—YEO_0.3% samples, respectively. As previously explained, despite drastic changes to pH, TVB-N, and TBARs in the control, these indicators were not significantly (*p* > 0.05) affected in the Ch_2%—YEO_0.3% sample, indicating that chitosan coating incorporating YEO was successful in preventing chicken meat spoilage during storage. The dual-sensors in the package headspace of the control sample successfully responded to the increase in pH value. There was a three-stage freshness status shown by concomitant changes in dual-sensors response and chemical indicators. The fresh stage in days 1–7 corresponded to non-significant (*p* > 0.05) changes in both sensor response and chemical indicators (Figure 6(A1–C1)). The semi-fresh stage on days 7–9 was attributed to the start of a sharp increase in sensor response and all chemical indicators. The values of 6.2, 19.8 mg N/100 g, and 0.45 mg MDA/kg obtained for pH, TVB-N, and TBARs on day 7 provided the onset of detection (shown by arrows) for dual-sensors to start color changing. These values were significantly (*p* < 0.05) increased to 6.8, 29.9 mg N/100 g, and 0.62 mg MDA/kg, respectively on day 9, showing a threshold of spoilage since no more changes occurred from this point onward. Thus, day 9 corresponded to the spoilage stage of uncoated chicken meat. Both sensors successfully followed a similar trend as shown by the pH, TVB-N, and TBARs response. It can be stated that as long as the chicken meat samples produce a similar amount of basic volatile compounds (TVB-N and/or TBARs) inside the headspace of the package, and thereby the dual-sensors will also give a similar response indicating a similar degree of freshness upon different storage times.

### 3.7. Response of Dual-Sensors to Total Aerobic Psycrotrophic Bacteria (APB) Enumeration

The effects of the application of different coating treatments and storage time on the viability of total aerobic mesophilic bacteria (AMB), aerobic psycrotrophic bacteria (APB); coliforms (*Enterobacteriaceae*), and yeast and molds were explained in Section 3.3 (Table 4). Here, to show the dependence of microbial change of meat samples in relation to dual-sensors response the variation of only APB count was plotted against mean RGB values of dual-sensors. Figure 7(A1,A2) demonstrates the color response of sensors along with APB numeration in control and Ch_2%—YEO_0.3% samples, respectively. APB and sensor label response (mean RGB values) were significantly (*p* < 0.05) affected in the control sample again showing a three-stage freshness status. It was shown that the dual-sensors in the package headspace of the control sample successfully and accurately responded to the increase in the psycrotrophic bacteria population in the meat. Whereas the Ch_2%—YEO_0.3% sample stayed fresh with no microbial growth, it also was confirmed by no color change of MR and BCP sensors. This shows that chitosan coating incorporating YEO successfully prevented chicken meat spoilage induced by APB enumeration during storage days. These results support the response of chemical indicators shown in Section 3.6, indicating that the increase in pH and TVB-N values probably originated from the growth and activity of psycrotrophic bacteria in chicken meat upon refrigerated storage time.

### 3.8. Response of Dual-Sensors to Visual Color of Chicken Meat

The variation in the color of chicken meat samples, as well as changes in dual-sensors label color, was qualitatively (visually) evaluated and the results were shown in Figure 2. To quantify these color variations, the response of dual-sensors toward the variation of color indices of uncoated and coated chicken meat was investigated. Figure 7(B1–C2) demonstrates the relation between sensors’ response with C* and WI of meat, respectively. Color indices of both meat samples (control and coated sample) were successfully related to the variation of sensor label response. Sensor label response in the control was significantly (*p* < 0.05) affected by storage time demonstrating a three-stage freshness status alongside a variation of C* (Figure 7(B1)) and WI (Figure 7(C1)). In the coated sample sensor response and meat color indices were not affected by storage time indicating that chitosan coating with YEO successfully prevented chicken meat spoilage. These results are in parallel with visual observation of meat color as shown in Figure 3, where a drastic color change in control (uncoated) meat occurred on day 9, whereas in Ch_2%—YEO_0.3% meat color was unchanged over all storage days. Moreover, the response of chemical indicators supports these results demonstrating that microbial development induces an increase in chemical indicators and consequently changes chicken meat color upon storage times concomitantly recognized by the sensor color.

## 4. Conclusions

In this study, a dual-sensors label was developed based on BCP and MR color indicators and used for monitoring the freshness quality of chicken breast fillets as functions of different coating types and chilled storage days. The dual-sensors successfully responded to the variation of chemical and microbiological indicators and visual color changes of different uncoated and coated samples during storage time. Control chicken meat showed a three-stage color change corresponding to different freshness statuses, where the fresh stage belonged to a period earlier than day 7 (with no distinct color change in both sensor labels); the semi-fresh stage corresponded to storage days 7–9, wherein a gradual color change appeared in sensor labels (MR from pink to orange, BCP from yellow to light purple); and the spoiled stage occurred in day 9 onward with a drastic color change (MR from orange to light yellow, BCP from light purple to deep purple). Samples coated with chitosan incorporating YEO did not show color changes in both sensors over all storage days indicating a single fresh status of chicken meat. Chitosan-based coatings in combination with YEO potentially result in extended shelf-life of chicken breast meat by decelerating microbial activities and chemical oxidation reactions due to the potential antioxidant and antimicrobial activities of YEO. In general, the dual-sensors label reacted accurately to meat freshness stages as shown by its reliability and sensitivity of detection in real-time monitoring of chicken meat freshness. The combination of food preservation techniques (i.e., active edible coating application) in extending the freshness of food with real-time spoilage monitoring serves as timely observing the degree of food spoilage with naked eyes while reducing food waste.

## Figures and Tables

**Figure 1 foods-11-03533-f001:**
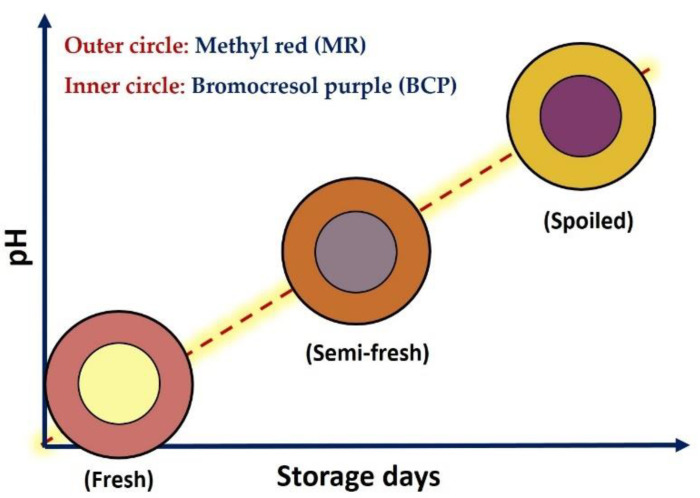
Color design of freshness dual-sensor labels based on the response of methyl red (MR, outer circle) and bromocresol purple (BCP, inner circle) upon pH variation as an indication of fresh, semi-fresh (should be consumed in hours) and spoiled (should not be consumed) chicken breast fillet.

**Figure 2 foods-11-03533-f002:**
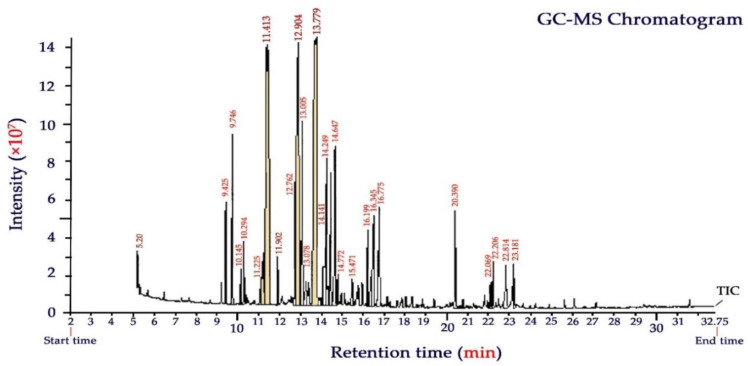
GC–MS chromatogram of Yarrow (*Achillea millefolium* L.) essential oil.

**Figure 3 foods-11-03533-f003:**
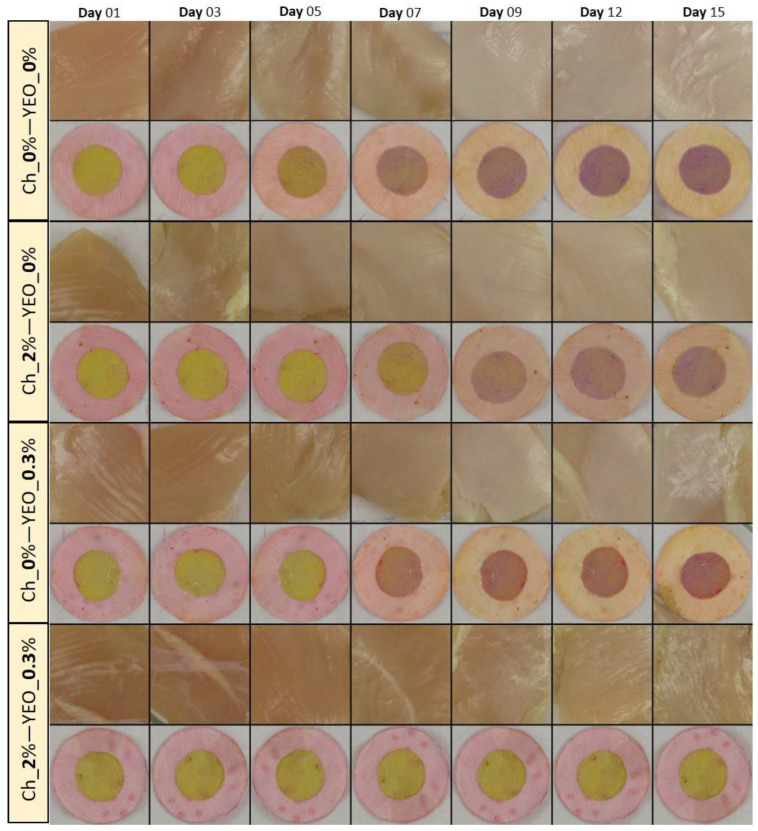
Visual inspection of the color of chicken meat samples and their attached sensor labels: effect of different types of coating and chilled storage time.

**Figure 4 foods-11-03533-f004:**
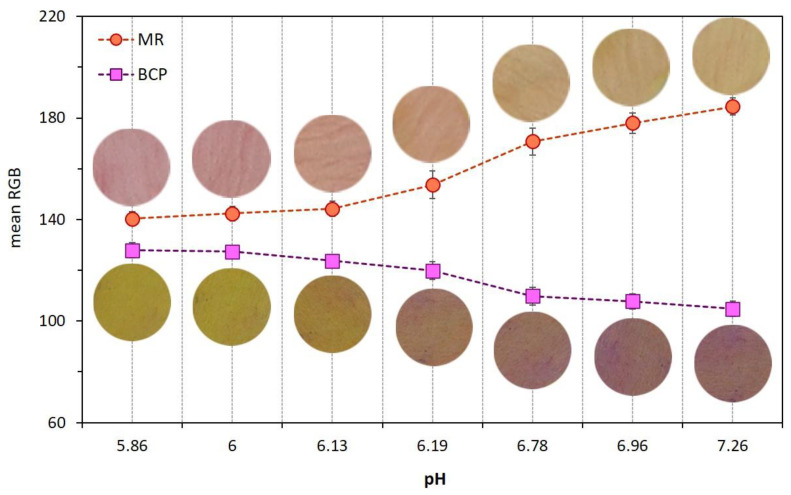
Effect of pH variation of uncoated chicken meat on dual-sensors label visual color and their mean RGB values.

**Figure 5 foods-11-03533-f005:**
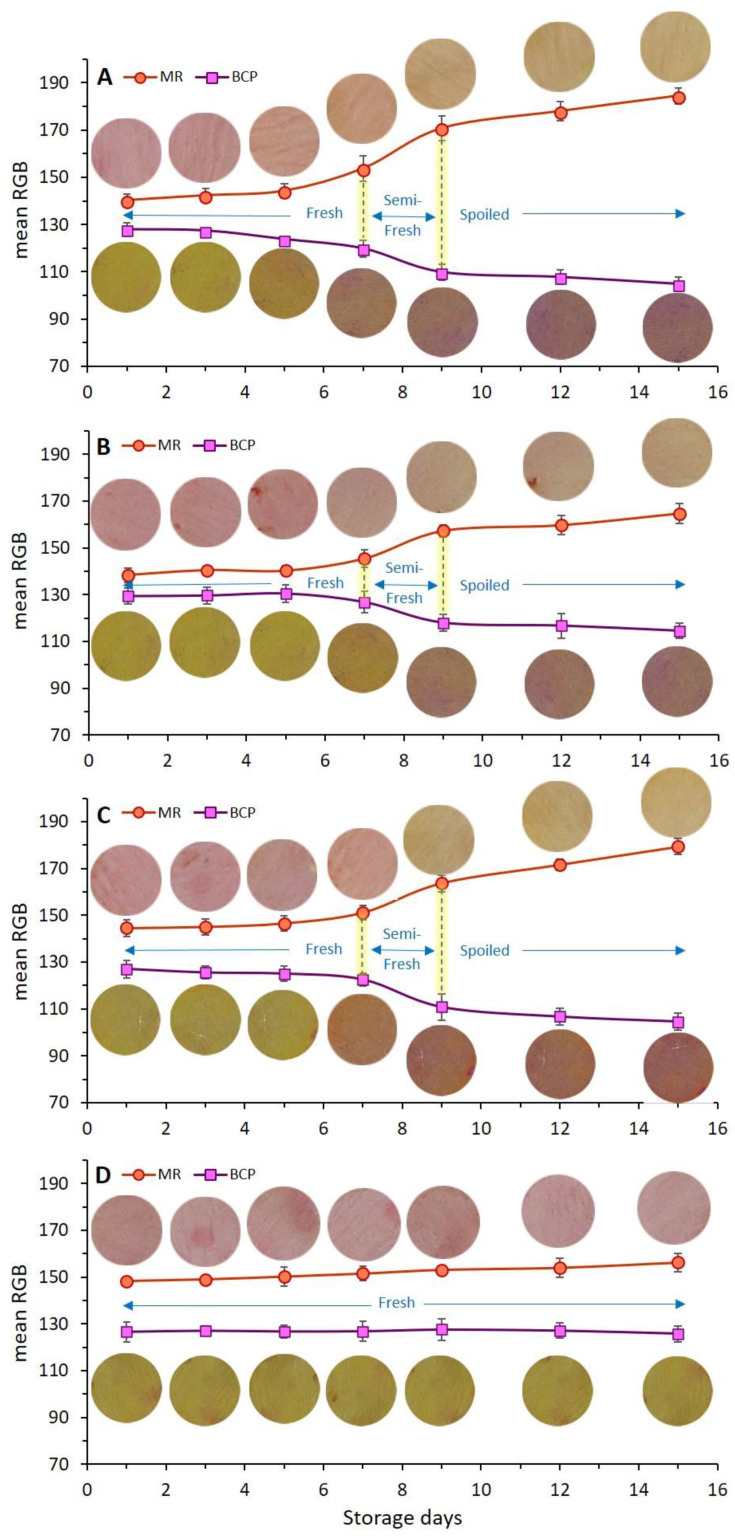
Effect of different types of chitosan coatings incorporating YEO ((**A**), control: Ch_0%—YEO_0%, (**B**): Ch_2%—YEO_0%, (**C**): Ch_0%—YEO_0.3%, (**D**): Ch_2%—YEO_0.3%) and chilled storage time on dual-sensors color responses (mean RGB) towards spoiling chicken breast fillets at chilled temperatures. Data are mean of at least triplicate measurements and error bars indicate SDs. MR: methyl red, BCP: Bromocresol purple. (For measurement of the mean RGB of each color sensor, the reader is referred to see Section 2.9 of this article).

**Figure 6 foods-11-03533-f006:**
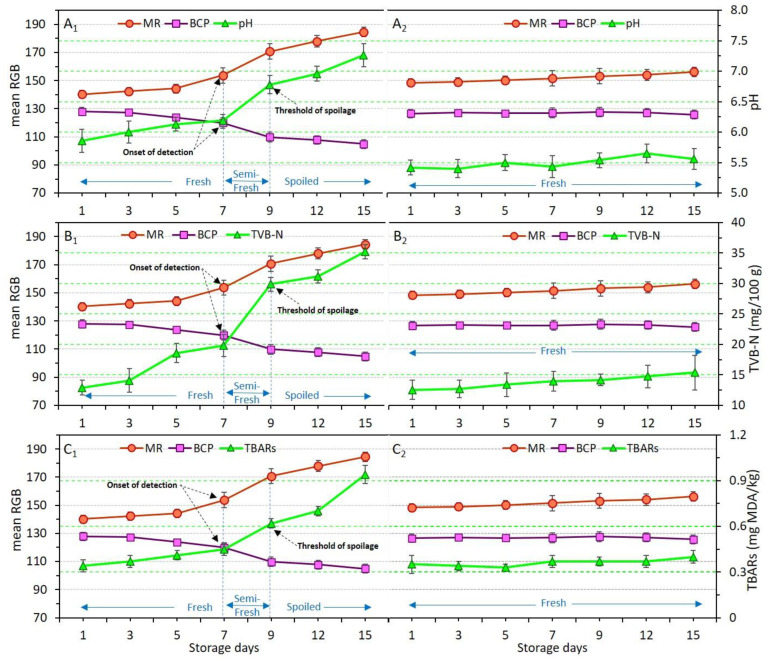
Variations of pH (**A**), TVB-N (**B**), and TBARs (**C**) values in relation to dual-sensors color responses (measured as mean RGB) in chicken meat samples without coating (control, (**A1**–**C1**) and coated with chitosan incorporating YEO (Ch_2%—YEO_0.3%, (**A2**–**C2**)) during chilled storage days. Data are mean of at least triplicate measurements and error bars indicate SDs. MR: methyl red, BCP: Bromocresol purple. Onset of detection and threshold of spoilage in relation to pH, TVB-N, and TBARs values has been shown by black arrows.

**Figure 7 foods-11-03533-f007:**
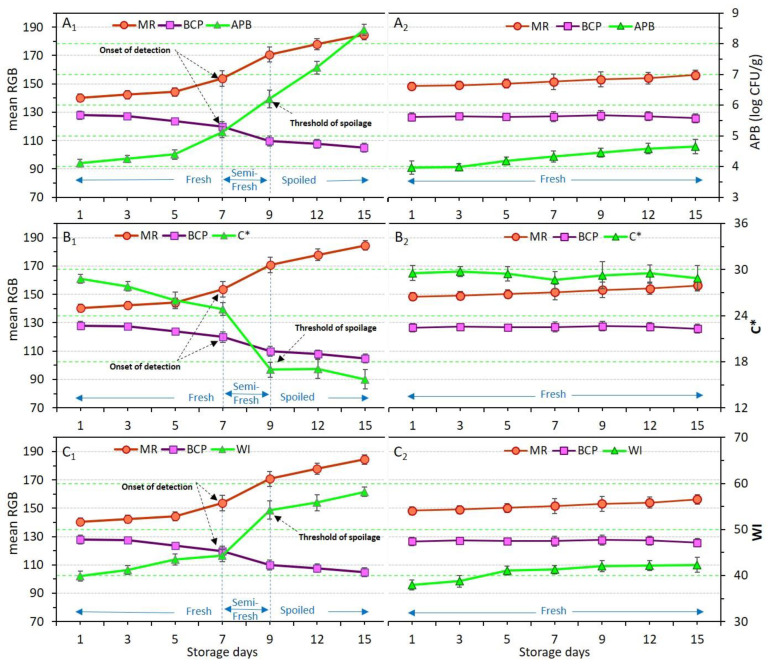
Variations of total aerobic psycrotrophic bacteria (APB) count (**A**), chroma index (C*) (**B**) and whiteness index (WI) (**C**) in relation to dual-sensors color responses (measured as mean RGB) in chicken meat samples without coating (control, (**A1**–**C1**)) and coated with chitosan containing YEO (Ch_2%—YEO_0.3%, (**A2**–**C2**)) during chilled storage days. Data are mean of at least triplicate measurements and error bars indicate SDs. MR: methyl red, BCP: Bromocresol purple. Onset of detection and threshold of spoilage in relation to APB, C*, and WI values has been shown by arrows.

**Table 1 foods-11-03533-t001:** Culture media, seeding techniques, and conditions of incubation used in microbiological characterization.

Microorganism Type	Culture Media	Seeding Technique	Incubation Period
Total AMB (aerobic mesophilic bacteria)	Plate count agar (PCA)	In depth	48 h/37 °C
Total APB (aerobic psycrotrophic bacteria)	Plate count agar (PCA)	In depth	7 days/7 °C
Coliforms (*Enterobacteriaceae*)	Violet red bile glucose (VRBG)	In depth/double layer	24 h/37 °C
Yeasts and Molds	Potato dextrose agar (PDA)	Surface	5 days/25 °C

**Table 2 foods-11-03533-t002:** Identified chemical components of Yarrow (*Achillea millefolium* L.) essential oil.

No	Retention Time (min)	KI ^a^	KI ^b^	Identified Compounds	Area ^c^ (%)
1	9.425	957	952	α-Pinene	1.49
2	9.746	963	953	Camphene	2.97
3	10.145	985	976	Sabinene	0.53
4	10.294	991	980	β-Pinene	0.97
5	11.117	1028	1026	*p*-Cymene	0.89
6	11.225	1033	1031	β-Ocimene	1.75
7	11.413	1042	1039	**Eucalyptol**	**17.07**
8	12.762	1102	1098	Linalool	3.57
9	12.904	1109	1110	**α-Thujone**	**12.10**
10	13.005	1114	1115	Cyclooctanone	1.60
11	13.078	1117	1115	**β-Thujone**	**3.81**
12	13.250	1126	1138	cis-Limonene oxide	1.44
13	13.779	1151	1143	**Camphor**	**18.55**
14	13.975	1160	1162	Pinocarvone	1.63
15	14.062	1164	1169	Lavandulol	1.14
16	14.249	1173	1165	Borneol	3.78
17	14.647	1192	1189	**α-Terpineol**	**7.23**
18	14.772	1198	1202	Myrtenol	0.51
19	15.471	1231	1242	Carvone	0.64
20	16.199	1269	1256	Neral (β-citral)	1.95
21	16.496	1284	1274	Geranial (α-citral)	2.51
22	16.755	1297	1294	Limonene dioxide	2.40
23	20.390	1493	1383	Geranyl acetate	1.65
24	22.069	1601	1583	Caryophyllene oxide	0.51
25	22.206	1609	1575	Spathulenol	0.79
26	22.814	1684	1525	1-Naphthalenol	1.23
27	23.181	1734	1652	α-Eudesmol	0.82
				Total identified compounds (%)	94.65

A DB-5 column (polydimethylsiloxane, dimensions: 60 m × 0.25 mm), a flame-ionized detector operating at electron ionization mode at 70 eV, and carrier gas of helium at a flow rate of 1.2 mL/min were used for the characterization of GC-MS components. The compounds are listed in order of their retention time on the DB-5 column. KI ^a^: Experimental Kovats indices calculated from the GC-MS chromatogram. KI ^b^: Kovats indices matching with those of published literature (NIST, Wiley). ^c^ Relative percentages of constituents using peak area normalization. (i.e., the peak areas for solvent compounds appeared earlier than the lowest number of n-alkanes (e.g., RT of 9.425 in this study) were discarded and the area of individual peaks was normalized).

**Table 3 foods-11-03533-t003:** Changes in pH, thiobarbituric acid reactive substances (TBARs), total volatile basic nitrogen (TVB-N) and pH values of different coating treatments of chicken breast fillets during different storage days at 4 °C.

	Treatments			Storage Days				
		1	3	5	7	9	12	15
pH	Ch_0%—YEO_0%	5.86 ^c,A^	6.00 ^c,A^	6.13 ^c,A^	6.19 ^c,A^	6.78 ^b,A^	6.96 ^b,A^	7.26 ^a,A^
		(0.19)	(0.18)	(0.11)	(0.10)	(0.15)	(0.12)	(0.19)
	Ch_2%—YEO_0%	5.33 ^d,B^	5.46 ^d,B^	5.55 ^d,B^	5.65 ^d,B^	6.17 ^c,B^	6.54 ^b,B^	6.89 ^a,B^
		(0.17)	(0.14)	(0.12)	(0.10)	(0.15)	(0.10)	(0.12)
	Ch_0%—YEO_0.3%	5.94 ^c,A^	6.05 ^c,A^	6.10 ^c,A^	6.21 ^c,A^	6.75 ^b,A^	6.86 ^b,A^	7.15 ^a,A^
		(0.11)	(0.15)	(0.13)	(0.11)	(0.12)	(0.09)	(0.13)
	Ch_2%—YEO_0.3%	5.42 ^b,C^	5.40 ^b,C^	5.51 ^ab,C^	5.43 ^a,C^	5.54 ^a,C^	5.65 ^a,C^	5.56 ^a,C^
		(0.12)	(0.15)	(0.13)	(0.18)	(0.12)	(0.15)	(0.17)
TBARs (mg MDA/kg)	Ch_0%—YEO_0%	0.34 ^d,A^	0.37 ^d,A^	0.41 ^d,A^	0.45 ^d,A^	0.62 ^c,A^	0.70 ^b,A^	0.94 ^a,A^
		(0.04)	(0.04)	(0.03)	(0.04)	(0.03)	(0.03)	(0.06)
	Ch_2%—YEO_0%	0.33 ^d,A^	0.36 ^d,A^	0.38 ^d,A^	0.42 ^d,A^	0.53 ^c,B^	0.60 ^b,B^	0.71 ^a,B^
		(0.03)	(0.03)	(0.04)	(0.03)	(0.04)	(0.03)	(0.06)
	Ch_0%—YEO_0.3%	0.36 ^d,A^	0.37 ^d,A^	0.43 ^d,A^	0.45 ^d,A^	0.59 ^c,B^	0.69 ^b,A^	0.89 ^a,A^
		(0.02)	(0.04)	(0.03)	(0.06)	(0.03)	(0.03)	(0.06)
	Ch_2%—YEO_0.3%	0.35 ^a,A^	0.34 ^a,A^	0.33 ^a,B^	0.37 ^a,B^	0.37 ^a,C^	0.37 ^a,C^	0.40 ^a,C^
		(0.06)	(0.03)	(0.02)	(0.04)	(0.03)	(0.04)	(0.04)
TVB-N (mg/100 g)	Ch_0%—YEO_0%	12.91 ^d,A^	14.06 ^d,A^	18.58 ^c,A^	19.82 ^c,A^	29.87 ^b,A^	31.12 ^b,A^	35.20 ^a,A^
		(1.25)	(1.93)	(1.55)	(1.82)	(1.13)	(1.08)	(1.14)
	Ch_2%—YEO_0%	11.89 ^d,A^	13.90 ^d,A^	17.52 ^c,A^	17.77 ^c,A^	27.50 ^b,A^	31.08 ^a,A^	33.51 ^a,B^
		(1.12)	(1.84)	(1.65)	(1.73)	(1.94)	(1.50)	(1.54)
	Ch_0%—YEO_0.3%	12.12 ^d,A^	14.12 ^d,A^	17.93 ^c,A^	18.21 ^c,A^	28.80 ^b,A^	32.53 ^a,A^	33.52 ^a,B^
		(0.94)	(1.24)	(1.33)	(1.20)	(1.90)	(2.13)	(1.62)
	Ch_2%—YEO_0.3%	12.55 ^a,A^	12.68 ^a,A^	13.40 ^a,B^	13.96 ^a,B^	14.15 ^a,B^	14.75 ^a,B^	15.36 ^a,C^
		(1.54)	(1.43)	(1.94)	(1.60)	(0.95)	(1.84)	(1.82)

Data are presented in mean (*n* = 3). SD values are given in parentheses below the means. Different lower-case and upper-case letters indicate significant (*p* < 0.05) differences between the means in the same rows and columns, respectively.

**Table 4 foods-11-03533-t004:** Effect of different types of coating and storage time on the population of total aerobic mesophilic bacteria (AMB), total aerobic psycrotrophic bacteria (APB), coliforms (*Enterobacteriaceae*), and yeasts and molds in chicken breast fillets stored at 4 °C.

	Treatments			Storage Days				
		1	3	5	7	9	12	15
AMB (log CFU/g)	Ch_0%—YEO_0%	4.04 ^e,A^	4.35 ^e,A^	4.58 ^e,A^	5.05 ^d,A^	6.13 ^c,A^	7.10 ^b,A^	8.52 ^a,A^
		(0.15)	(0.18)	(0.17)	(0.14)	(0.19)	(0.11)	(0.14)
	Ch_2%—YEO_0%	3.91 ^d,A^	4.17 ^d,B^	4.32 ^d,B^	4.41 ^d,C^	5.49 ^c,C^	6.54 ^b,C^	7.65 ^a,C^
		(0.10)	(0.11)	(0.14)	(0.08)	(0.10)	(0.11)	(0.12)
	Ch_0%—YEO_0.3%	4.10 ^e,A^	3.95 ^e,B^	4.18 ^e,B^	4.72 ^d,B^	6.25 ^c,B^	7.26 ^b,B^	8.36 ^a,B^
		(0.10)	(0.11)	(0.13)	(0.18)	(0.12)	(0.09)	(0.13)
	Ch_2%—YEO_0.3%	3.92 ^a,A^	4.20 ^a,B^	4.34 ^a,B^	4.47 ^a,C^	4.54 ^a,D^	4.60 ^a,D^	4.74 ^a,D^
		(0.13)	(0.14)	(0.12)	(0.11)	(0.16)	(0.10)	(0.14)
APB (log CFU/g)	Ch_0%—YEO_0%	4.11 ^d,A^	4.25 ^d,A^	4.39 ^d,A^	5.12 ^c,A^	6.20 ^b,A^	7.22 ^a,A^	8.45 ^a,A^
		(0.13)	(0.12)	(0.16)	(0.18)	(0.28)	(0.21)	(0.18)
	Ch_2%—YEO_0%	3.99 ^d,A^	3.90 ^d,B^	4.15 ^e,C^	4.70 ^d,C^	5.55 ^c,C^	6.34 ^b,C^	7.21 ^a,C^
		(0.10)	(0.08)	(0.14)	(0.09)	(0.12)	(0.10)	(0.17)
	Ch_0%—YEO_0.3%	4.05 ^e,A^	3.90 ^e,A^	3.96 ^e,B^	4.36 ^d,B^	6.00 ^c,B^	7.21 ^b,B^	7.98 ^a,B^
		(0.07)	(0.14)	(0.12)	(0.18)	(0.11)	(0.09)	(0.18)
	Ch_2%—YEO_0.3%	3.97 ^a,A^	3.98 ^a,B^	4.18 ^a,C^	4.32 ^a,D^	4.45 ^a,D^	4.58 ^a,D^	4.65 ^a,D^
		(0.22)	(0.12)	(0.13)	(0.18)	(0.14)	(0.17)	(0.23)
Coliforms (*Enterobacteriaceae*) (log CFU/g)	Ch_0%—YEO_0%	3.89 ^e,A^	4.05 ^e,A^	4.29 ^d,A^	4.65 ^d,A^	6.21 ^c,A^	7.22 ^b,A^	7.95 ^a,A^
		(0.18)	(0.19)	(0.21)	(0.18)	(0.18)	(0.15)	(0.13)
	Ch_2%—YEO_0%	3.75 ^d,A^	3.89 ^d,B^	3.95 ^d,B^	4.26 ^d,B^	5.96 ^c,C^	7.12 ^b,C^	7.90 ^a,C^
		(0.10)	(0.14)	(0.19)	(0.15)	(0.11)	(0.12)	(0.15)
	Ch_0%—YEO_0.3%	3.26 ^e,A^	3.35 ^e,B^	3.25 ^e,C^	3.76 ^d,A^	5.75 ^c,B^	7.43 ^b,B^	8.50 ^a,B^
		(0.17)	(0.13)	(0.11)	(0.17)	(0.10)	(0.10)	(0.16)
	Ch_2%—YEO_0.3%	3.59 ^a,A^	3.68 ^a,B^	3.75 ^a,C^	3.82 ^a,C^	3.98 ^a,D^	4.15 ^a,D^	4.25 ^a,D^
		(0.10)	(0.14)	(0.10)	(0.20)	(0.14)	(0.16)	(0.15)
Yeast and molds (log CFU/g)	Ch_0%—YEO_0%	3.10 ^d,A^	3.24 ^d,A^	3.42 ^c,A^	3.65 ^c,A^	6.47 ^b,A^	7.12 ^b,A^	8.20 ^a,A^
		(0.09)	(0.10)	(0.11)	(0.21)	(0.13)	(0.20)	(0.11)
	Ch_2%—YEO_0%	2.94 ^e,A^	3.09 ^e,B^	3.29 ^e,B^	3.45 ^d,B^	5.94 ^c,B^	6.51 ^b,B^	7.11 ^a,B^
		(0.07)	(0.08)	(0.11)	(0.05)	(0.09)	(0.15)	(0.18)
	Ch_0%—YEO_0.3%	3.00 ^e,A^	3.18 ^e,B^	3.25 ^e,B^	4.11 ^d,A^	6.50 ^c,A^	6.95 ^b,A^	7.70 ^a,A^
		(0.07)	(0.10)	(0.11)	(0.12)	(0.09)	(0.07)	(0.16)
	Ch_2%—YEO_0.3%	3.00 ^a,A^	3.12 ^a,C^	3.20 ^a,C^	3.26 ^a,C^	3.35 ^a,C^	3.40 ^a,C^	3.66 ^a,C^
		(0.15)	(0.14)	(0.19)	(0.14)	(0.19)	(0.13)	(0.14)

Data are presented in mean (*n* = 3). SD values are given in parentheses below the means. Different lower-case and upper-case letters indicate significant (*p* < 0.05) differences between the means in the same rows and columns, respectively.

**Table 5 foods-11-03533-t005:** Effect of different types of coating and storage time on color attributes (C* = chroma, WI = whiteness index, YI = yellowness index) of chicken breast fillets stored at 4 °C.

	Treatments			Storage Days				
		1	3	5	7	9	12	15
Chroma (C*)	Ch_0%—YEO_0%	28.80 ^a,A^	27.81 ^a,A^	25.98 ^a,A^	24.85 ^a,B^	16.98 ^b,C^	17.05 ^b,C^	15.71 ^b,C^
		(0.55)	(0.62)	(1.07)	(0.82)	(0.97)	(1.23)	(1.27)
	Ch_2%—YEO_0%	32.19 ^a,A^	31.33 ^a,A^	25.78 ^b,A^	25.31 ^b,B^	19.53 ^c,B^	19.63 ^c,B^	19.92 ^c,B^
		(2.32)	(0.95)	(2.75)	(1.86)	(1.13)	(0.43)	(1.42)
	Ch_0%—YEO_0.3%	30.39 ^a,A^	30.38 ^a,A^	29.10 ^a,A^	25.31 ^b,B^	19.29 ^c,B^	16.63 ^d,C^	16.65 ^d,C^
		(1.12)	(0.69)	(1.23)	(1.00)	(0.58)	(0.43)	(1.96)
	Ch_2%—YEO_0.3%	29.55 ^a,A^	29.74 ^a,A^	29.43 ^a,A^	28.63 ^a,A^	29.25 ^a,A^	29.49 ^a,A^	28.90 ^a,A^
		(0.99)	(0.68)	(0.93)	(1.11)	(1.76)	(1.10)	(1.65)
Whiteness index (WI)	Ch_0%—YEO_0%	39.90 ^b,A^	41.25 ^b,A^	43.47 ^b,A^	44.34 ^b,A^	54.22 ^a,A^	55.82 ^a,A^	58.17 ^a,A^
		(1.10)	(0.98)	(1.2)	(1.28)	(1.95)	(1.71)	(1.04)
	Ch_2%—YEO_0%	41.34 ^c,A^	43.71 ^b,A^	44.54 ^b,A^	49.21 ^a,A^	51.61 ^a,A^	51.81 ^a,A^	50.77 ^a,A^
		(0.82)	(0.40)	(0.95)	(1.11)	(2.66)	(2.06)	(2.08)
	Ch_0%—YEO_0.3%	39.33 ^c,A^	41.05 ^c,A^	42.99 ^c,A^	46.61 ^c,A^	53.33 ^b,A^	53.19 ^b,A^	57.79 ^a,A^
		(3.46)	(1.57)	(1.26)	(1.63)	(1.39)	(0.57)	(1.72)
	Ch_2%—YEO_0.3%	38.00 ^a,A^	38.57 ^a,A^	41.12 ^a,A^	41.29 ^a,A^	42.05 ^a,A^	42.16 ^a,A^	42.35 ^a,A^
		(1.07)	(1.25)	(0.99)	(0.91)	(1.24)	(1.13)	(1.59)
Yellowness index (YI)	Ch_0%—YEO_0%	80.14 ^a,A^	76.24 ^a,A^	69.99 ^b,B^	66.46 ^b,B^	40.99 ^c,C^	39.79 ^c,D^	36.14 ^c,C^
		(2.33)	(1.43)	(2.65)	(2.42)	(2.26)	(3.22)	(2.73)
	Ch_2%—YEO_0%	86.08 ^a,A^	80.48 ^a,A^	67.79 ^b,B^	61.21 ^b,B^	48.80 ^c,B^	49.18 ^c,B^	51.34 ^c,B^
		(5.42)	(2.74)	(4.21)	(4.08)	(4.26)	(2.86)	(4.13)
	Ch_0%—YEO_0.3%	84.59 ^a,A^	80.89 ^a,A^	77.07 ^a,A^	64.09 ^b,B^	46.62 ^c,B^	41.28 ^c,C^	38.52 ^c,C^
		(7.23)	(3.23)	(3.75)	(4.10)	(2.13)	(1.44)	(4.78)
	Ch_2%—YEO_0.3%	86.55 ^a,A^	86.50 ^a,A^	80.94 ^a,A^	79.11 ^a,A^	79.33 ^a,A^	79.65 ^a,A^	74.75 ^a,A^
		(7.05)	(2.70)	(3.56)	(2.98)	(5.45)	(3.73)	(8.28)

Data are presented in mean (*n* = 3). SD values are given in parentheses below the means. Different lower-case and upper-case letters indicate significant (*p* < 0.05) differences between the means in the same rows and columns, respectively.

## Data Availability

Data is contained within the article.

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
