# Peer review of "Development and Application of Dual-Sensors Label in Combination with Active Chitosan-Based Coating Incorporating Yarrow Essential Oil for Freshness Monitoring and Shelf-Life Extension of Chicken Fillet"

_foods, 2022, doi:10.3390/foods11213533_

Round 1

Reviewer 1 Report

The article entitled “application of active chitosan coating incorporating yarrow essential oil in combination with freshness detecting sensor for shelf life monitoring and extending of chicken breast fillet” was well written and the results was comprehensively discussed.

Title of the article needs to revised. How is the shelf life monitored? In this case, the freshess is monitored, not the shelf life. 

Line 106-107, the objective of the study is “to develop and characterize intelligent on-package dual-sensors label…..”, but according to the results, it did show characterization of the label, but it tended to report the effects of label to the chicken breast. 

Besides, the good results, pH responsive colour of film with different pH was not reported, which is important to see how did film change in different pH solution. With this report, we can see the potential application of the film for other food products.

Author Response

Response to Reviewer 1 is attached

Reviewer 2 Report

In this manuscript, authors have developed an active chitosan coating incorporated with Yarrow essential oil, then the coating was used to maintain the freshness of chicken breast fillet, while the shelf-life of chicken breast fillet was monitored by on-package sensor label based on bromocresol purple (BCP) and methyl red (MR). This work was well organized, and a large number of experiments were conducted and some interesting results were obtained. However, the novelty and scientific significance of this work is poor, and there are some crucial and essential issues and concerns should be improved and clarified, as below.

1. Polysaccharides have hydrophilic property, whether the chitosan-based coating will become sticky (weak mechanical strength) because of the absorption of external water in the process of food preservation, so the loss of protection ability.

2. How to ensure that hydrophilic chitosan and hydrophobic essential oil can evenly emulsify without adding surfactant to form a coating that can protect the chicken breast fillet?

3. How about the biosafety of BCP and MR? Are they allowed to directly add or come into contact with food? The reviewer worried that BCP and MR may enter the protected food in the author's experimental design, which would cause food safety problems.

4. Dual circle-sensors stickers were positioned close to meat samples inside packaging atmosphere. Hence, how can the sensor quickly detect the volatile basic amines produced by chicken spoilage, because the volatile basic amines can only be detected after it passes through the chitosan-based coating. How about the permeability of chitosan coating?

5. The authors emphasized that pH and TBARs were within the range of acceptable values indicating no microbial spoilage due to antibacterial and antioxidant properties of chitosan coating incorporating YEO. What’s the range of acceptable values of pH and TBARs? Moreover, the authors should provide the quantized antibacterial and antioxidant properties of chitosan-based coating.

6. Line 384-386, How did the authors get this conclusion based on the above experimental results?

7. Whether the active ingredients in the Yarrow essential oil will diffuse into the chicken breast fillet, thus affecting its flavor?

8. Line 581-587, the authors indicate that the dual-sensors label composed with two indicator dyes can serve as intelligent shelf-life labeling devices. In my opinion, the innovation of this paper lies in the combination of food preservation (extending the freshness of food) and real-time spoilage monitoring, so as to timely observe the degree of food spoilage with naked eyes while reducing food waste. The Conclusion section should be revised.

Author Response

Detailed response to Reviewer 2 comments is attached.

Round 2

Reviewer 2 Report

No further comment. Accept in present form.